# Morphology Determines an Efficient Coherent Electron Transport for Push–Pull Organic Semiconductors Based on Triphenylamine and Dicyanovinyl Groups

**DOI:** 10.3390/ma16062442

**Published:** 2023-03-18

**Authors:** Alexander Romero, Jaime Velasco-Medina, Alejandro Ortiz

**Affiliations:** 1Grupo de Bionanoelectrónica, Universidad del Valle, Calle 13 # 100-00, Cali 760001, Colombia; gilberto.zapata@correounivalle.edu.co; 2Grupo de Investigación de Compuestos Heterocíclicos, Universidad del Valle, Calle 13 # 100-00, Cali 760001, Colombia; alejandro.ortiz@correounivalle.edu.co

**Keywords:** organic semiconductor, charge transport, Landauer–Büttiker theory

## Abstract

The morphology of the active layer in organic solar cells is fundamental for achieving high power conversion efficiency. However, the morphological characteristics for optimal performance are still being investigated. An atomistic computational approach is required to determine the relationship between active layer morphology and performance. Since the organic solar cell has multiple phases and interfaces, the computational modeling of charge generation and transport is challenging. We then used a set of push–pull semiconductors to illustrate how the electronic transmission spectrum, derived from the Landauer–Büttiker formalism, can be used to investigate the efficiency of coherent charge transport across anisotropic organic solids. The electronic transmission spectrum was calculated from the electronic band structure obtained using the density-functional-based tight-binding method. We found that coherent charge transport was more efficient along the direction parallel with the interface between the electron-acceptor and electron-donor moieties for a herringbone morphology.

## 1. Introduction

The performance of organic solar cells cannot be easily predicted from the properties of single molecules [1,2]. The above is due to the intermolecular arrangement of the molecules that compose the active layer of organic solar cells. This arrangement affects the exciton diffusion and the charge generation and recombination [3,4,5,6]. The morphology of the active layer then influences the performance of organic solar cells. The morphology can be classified as global and local. The first is characterized by the domains’ size, purity, connectivity, and crystallinity. The second is related to the intermolecular arrangement or molecular packing in the interfaces and the bulk of the domains [7,8,9]. However, the characteristics of active layer morphology for optimal performance are not known yet [2]. Therefore, understanding the relationship between morphology and the efficiency of organic solar cells is required to improve the performance of organic solar cells [10]. Experimental techniques do not provide a complete description of molecular packing; computational methods complement the study of the relationship among chemical structure, morphology, and device performance [6].

In general, the standard transport theory for macroscopic bulk semiconductors is not appropriate for modeling the photovoltaics of nanostructure-based devices in which dimensional and quantum effects may have a significant role [11,12]. In this case, the Landauer–Büttiker theory could be used. Significant advances in understanding electron quantum transport have been achieved using this theory [13]. The Landauer–Büttiker theory describes quantum transport in devices with a semiclassical or quantum behavior, i.e., devices with lengths greater than or around the electron’s de Broglie wavelength. In both cases, the electron transport is supposed to be coherent and the mean-field description to be applicable. The above theory solves a single-particle scattering problem and determines the conductance using the transmission matrix coefficients. The coefficients of this matrix are the probabilities of electron transmission along the transport direction for each transport channel, which is determined by the transverse direction. The sum of the transmission matrix coefficients, which depends on the energy, is known as the electronic transmission function or spectrum (not related to spectroscopy). The electrical current is proportional to the average of the electronic transmission spectrum over the energy, weighted by the difference between the Fermi–Dirac distributions of the electrodes [14].

Under bias, the electronic transmission spectrum is usually calculated by using the non-equilibrium Green’s function formalism and the density functional theory using a device model that involves open boundary conditions [12,13,14]. However, the electronic transmission spectrum can be calculated from the electronic band structure for pure phases without bias using a simulation box with periodic boundary conditions. We used this last approach to investigate how the electrical conduction in an organic semiconductor depended on the transport direction. As a model, we used compounds **1** through **4**, whose structural formulas are shown in Figure 1. These compounds are push–pull semiconductors composed of a triphenylamine (electron-donor) and dicyanovinyl (electron-acceptor) group. The above compounds were used as electron-donor materials in planar bilayer organic solar cells [15].

## 2. Materials and Methods

In order to build structural models of the solid state of compounds **1** through **4**, we initially optimized the crystal structure reported in [15] by minimizing the interatomic forces and total stress. We then deformed the unit cells to build orthorhombic cells, the coordinate vectors of the atoms were modified according to the transformation of the lattice vectors. Finally, the orthorhombic structure was optimized, keeping the orthorhombic geometry.

The manipulation and visualization of the crystal structures and the simulations were done in the QuantumATK software suite (version S-2021.06) [16]. The electronic state was calculated using: (1) the density-functional-based tight-binding method, with mio-1-1 parameters [17,18]; (2) the Pulay mixing algorithm with a 10^−5^ E_h_ convergence tolerance; (3) the fast-Fourier-transform-based Poisson solver; (4) a Fermi–Dirac smearing scheme with a broadening corresponding to 300 K. The geometry optimization was done using: (1) a mesh cut-off of 100 E_h_; (2) a Monkhorst–Pack grid with a k-point density of ca. 10 Å; (3) a limited-memory Broyden-Fletcher-Goldfarb-Shanno algorithm; (4) a maximum force and stress of 0.05 eV Å^−1^ and 0.1 GPa as convergence parameters.

We calculated the band gap and Fermi level using different k-point numbers and mesh cut-off values. The k-point numbers and mesh cut-off values that were used are in Appendix A. The band gap of the bulk material was calculated from the electronic band structure, which was calculated along the k-point path Y-Γ-X-S-R-U-X-S-Y-T-Z-Γ-U-Z by using 20 k-points between each pair of high symmetry points.

Finally, we computed the electronic transmission spectrum under zero bias for energies between 2 eV around the Fermi level, considering different transport directions. The energy range was discretized using 201 points.

## 3. Results

The band gap and Fermi level were calculated for five sets of k-points. The sets correspond to the k-point densities of ca. 9, 16, 23, 30, and 52 Å along each reciprocal lattice vector. For each k-point set, we set the mesh cut-off to 40, 80, 100, 160, and 200 E_h_. We found that the band gap (up to one decimal digit) did not depend on the k-point number or mesh cut-off. The Fermi level was also independent of the mesh cut-off. Regarding the k-point number, only compound **1** showed minor variations (~0.1 eV) in the Fermi level. The obtained results are in Appendix A.

We computed the electronic transmission spectrum for three perpendicular directions: P, P1, and P2 (Figure 2). We found that the electronic transmission was negligible along the direction P for all compounds. Figure 2 shows the electronic transmission spectrum along the directions P1 and P2, calculated using a k-point density of ~9 Å. The band structure from which the spectrum was calculated was computed using a k-point density of ~52 Å. The electronic transmission was normalized with respect to the maximum value achieved by the compounds. This value corresponded to one of the transmission spectrum peaks along the direction P1 of compound **4**.

## 4. Discussion

The electronic transmission spectrum depends on the k-point sampling, making it challenging to obtain converged quantities that involve averaging over the spectrum [19,20]. However, the general behavior of the transmission spectrum can be obtained with a small number of k-points [21]. In order to evaluate how the spectrum depended on the k-point sampling, we compared the transmission spectrum using a k-point density of ca. 9 and 52 Å along each reciprocal lattice vector. We found the position and relative height of the spectrum peaks were roughly the same (see Appendix A). In this study, we used the electronic transmission spectrum to compare the coherent charge transport in the compounds qualitatively. We then did not need a highly converged electronic transmission spectrum but only needed to converge the main features of the spectrum. These features were the approximate position and relative height of the spectrum peaks. In this regard, a density of ~9 Å was enough.

The electronic transmission spectrum is computed from the electronic band structure; thus, this structure also had to be converged. The convergence of the band structure needed to be checked, taking into account the k-point sampling and mesh cut-off. To avoid computing the transmission spectrum for different values of mesh cut-off and k-point number, we calculated the band gap and Fermi level. We found only a variation in the Fermi level with respect to the number of k-point for compound **1**. We then compared the electronic transmission spectrum calculated from the band structure obtained with the low and high k-point densities (see Appendix A).

Except for compound **1**, the electronic band structure calculated with a k-point density of ca. 9 and 52 Å led to the same electronic transmission spectrum approximately. For compound **1**, there was a shift in the spectrum toward lower energies. We then extended the analysis by including higher k-point densities to calculate the band structure to test convergence. We used densities of ~92, 100, 105, 110, 116 Å and a 40 E_h_ mesh cut-off (see Appendix A). Except for the densities of ca. 105 and 116 Å, the Fermi level was like the one obtained using the density of ~52 Å. We then computed the transmission spectrum using the densities of ca. 110 and 116 Å to verify the spectrum convergence. These densities corresponded to the higher densities; the first represented the behavior of the majority, and the second represented the outliers. Contrary to 116 Å, the spectrum calculated using a density of ca. 110 Å was the same as when a density of ca. 52 Å was used (Figure 3). The variation of the Fermi level then correlated to a shift in the transmission spectrum relative to the Fermi level, which was the case for densities ca. 105 and 116 Å.

The number of energy levels in the conduction and valence bands will affect the Fermi level position. Moreover, the electronic transmission spectrum may change whether certain energy levels are considered or not for its calculation. Adding more k-points will affect the Fermi level and the transmission spectrum by including new energy levels. Therefore, the relative energy of the Fermi level with respect to the band edges must be converged before computing the transmission spectrum. A large number of k-points sets should be analyzed because the convergence is not variational. In our study, a converged transmission spectrum was achieved with a density of ~52 Å.

The electronic states nearest to the Fermi level contribute the most to the electrical current within the context of the Landauer–Büttiker formalism [22]. The transmission spectrum peaks nearest to the Fermi level corresponded to the energies in (−1.0, 1.0) eV (Figure 2). By analyzing these peaks, we found that electronic transmission was affected by the transport direction. The intermolecular arrangement within a solid phase of anisotropic molecules caused anisotropy in the solid. Different values of the charge transport properties along different directions in the solid were then expected. The crystal structures of compounds **1** to **4** showed two delimited regions corresponding to a phase of electron-donor moieties, next to a phase of electron-acceptor moieties separated by covalent and noncovalent interactions; this generated a periodic structure of alternating layers of electron-donor and electron-acceptor moieties (Figure 4). We then used the interfaces between these alternating layers to define the transport directions. We calculated the spectrum along a transport direction perpendicular (P) to and parallel (P1 and P2) with the interfaces.

We used the molecular plane described by the dicyanovinyl moiety to define the face-, end- and edge-on directions, such as the shortest, longest, and mid-length principal axes of molecules that were stacked approximately collinear [2]. The direction P, perpendicular to the sequential interfaces, corresponded to alternating end- and face-on stacking modes. The molecules showed an edge-on stacking along the direction P1. Along the direction P2, stacking was face-on except for compound **4**, whose molecular stacking was a mix between end- and face-on (Figure 4).

The Fermi level for each compound was into the band gap. Thus, in Figure 2, the peaks at positive energies correspond to the electron transport, and the peaks at negative energies correspond to the hole transport. Except for compound **4**, every compound showed a similar hole- and electron-transport efficiency since the height of the peaks at positive and negative energies was similar.

Compounds **1** and **3** showed a small difference between the electronic transmission spectra along the directions P1 and P2. For compound **1**, the electron transport along the direction P1 was slightly more efficient because the peaks were higher. Moreover, the transport along the direction P2 slightly favored the electron transport because the peak at negative energy was lower than the peak at positive energy. For compound **3**, electron transport was also favored. In this case, there was an additional peak at ~0.5 eV compared with that of the spectrum along the direction P1.

Compounds **2** and **4** showed a big difference when the transport directions P1 and P2 were compared. Along the direction P2, the coherent electron transport was better for compound **2**. Along the direction P1, the coherent electron transport was better for compound **4**, which among all the compounds was the one that had the highest transmission peaks. Along the direction P2, compound **4** transported electrons more efficiently than holes because the peak at the positive energy was higher.

We found that all compounds’ charge transmission was negligible along the direction P. Along this direction, interfaces separated the dicyanovinyl and phenylamine moieties. An interface parallel to the transport direction may be then necessary for efficient charge transport. However, the direction P1 was parallel with the interface, and the charge transmission was negligible for compound **2**. Therefore, there was another factor involved in achieving an efficient transport.

The charge transmission of compounds **1** and **3** along the direction P1 was around half that of the transmission of compound **4**. All the compounds in this direction had edge-on stacking, but they differed in the orientation of the molecular plane with respect to the interface. The angle between the molecular plane and the interface was ~90° for compound **2** and ~45° for compounds **1**, **3**, and **4**. Efficient charge transmission was related to a suitable distance and the orientation between the molecules composing the solid. The structural arrangement of compound **2** then seemed to reduce charge transmission in the direction P1. The higher charge transmission of compound **4** with respect to compounds **1** through **3** may be due to its herringbone arrangement (Figure 4). It was previously mentioned that a herringbone structure improves electrical conduction [23]. In this study, intermolecular charge transfer was enhanced using two-dimensional spatial paths.

Currently, there is no consensus about which molecular orientation leads to a more efficient charge generation when comparing edge- and face-on stacking modes [5,24]. We found that coherent charge transmission was higher along the edge-on stacking direction (P1) of compound **4**. However, this was not the case for the other compounds. A herringbone structure perpendicular to the transport direction may be necessary to achieve a high charge transmission. Moreover, due to the push–pull structure of the semiconductors, the interface between the electron-donor and electron-acceptor moieties may also contribute to an efficient coherent charge transport, where the charge transport direction should be parallel to the interface.

## 5. Conclusions

In [15], the power conversion efficiencies measured for planar bilayer organic solar cells, composed of ITO|PEDOT:PSS|electron-donor material|C60|Al, were 2.53, 2.97, 0.57, and 1.35% for compounds **1**, **2**, **3**, and **4**, respectively. However, we found a higher coherent charge transmission in compound **4**. For organic solar cells, the free-charge transport must be efficient across electron-donor and electron-acceptor phases, and in some cases, this transport limits the overall performance [25,26]. However, the free-charge transport is not the only process occurring in the photovoltaics phenomena. Here, we exemplified how the Landauer–Büttiker formalism could be used to study charge transmission in pure phases, considering the morphology of the phase. The measured efficiencies for compounds **1** to **4** were then not necessarily determined by the coherent charge transport through the electron-donor phase of the organic solar cell, which was the object of our study. Furthermore, our methodology has two main shortcomings that need to be addressed in future investigations to contribute substantially the advancement of organic solar cell technologies. First, the Landauer–Büttiker theory uses equilibrium-state electronic-structure methods, treats the system as open, and does not include decoherence effects. All these aspects are currently under investigation, leading to multiple extensions of the theory [13]. Second, the structural model was very simple because the experimental construction of the cell will lead to phases involving different crystalline and amorphous structures. This will heavily impact the charge transport, as shown here when comparing different transport directions for the same material. Therefore, we think the methodology used to build the structural model is not appropriate for predicting the efficiency of a cell. However, by providing an optimal morphology to target, this methodology is valuable in orienting the development of experimental techniques designed to control morphology.

## Figures and Tables

**Figure 1 materials-16-02442-f001:**
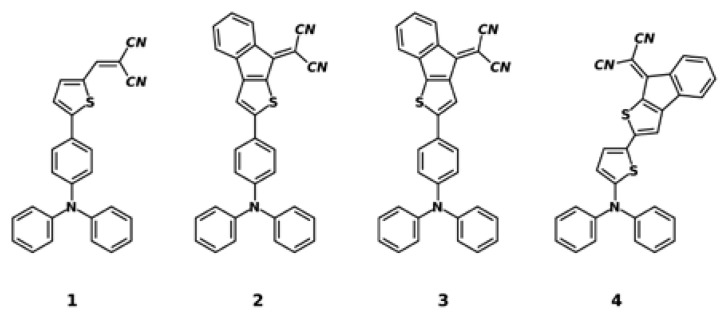
Structural formula of compounds **1**, **2**, **3**, and **4**.

**Figure 2 materials-16-02442-f002:**
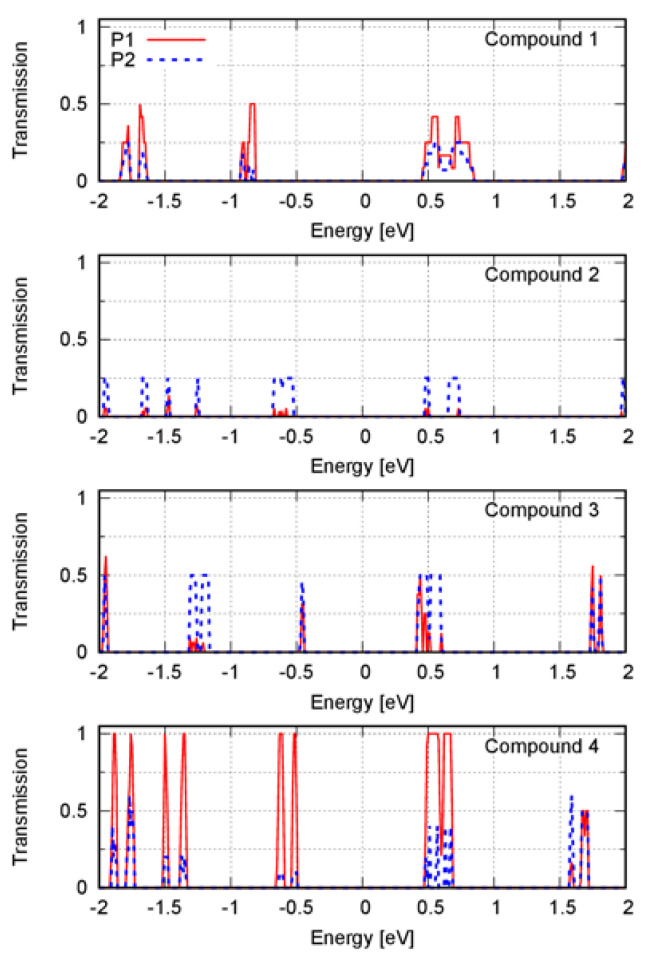
Electronic transmission spectrum along the transport directions P1 (solid red line) and P2 (dashed blue line) for compounds **1** through **4**. The energy was relative to the Fermi level, which was set to zero.

**Figure 3 materials-16-02442-f003:**
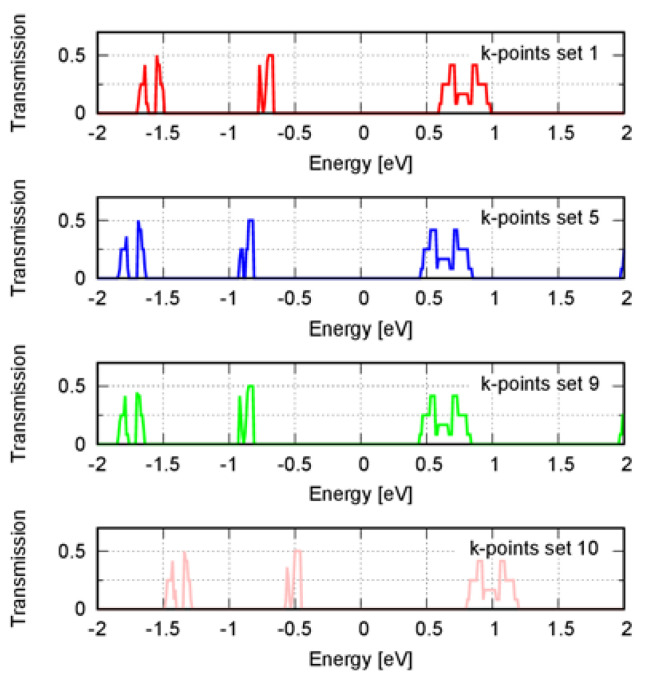
Electronic transmission spectrum along the transport direction P1 for compound **1** using a band structure calculated with different k-point numbers. Sets 1, 5, 9, and 10 corresponded to the k-point densities of ~9, 52, 110, and 116 Å, respectively. The transmission spectrum along the other directions is shown in Appendix A. The energy was relative to the Fermi level, which was set to zero.

**Figure 4 materials-16-02442-f004:**
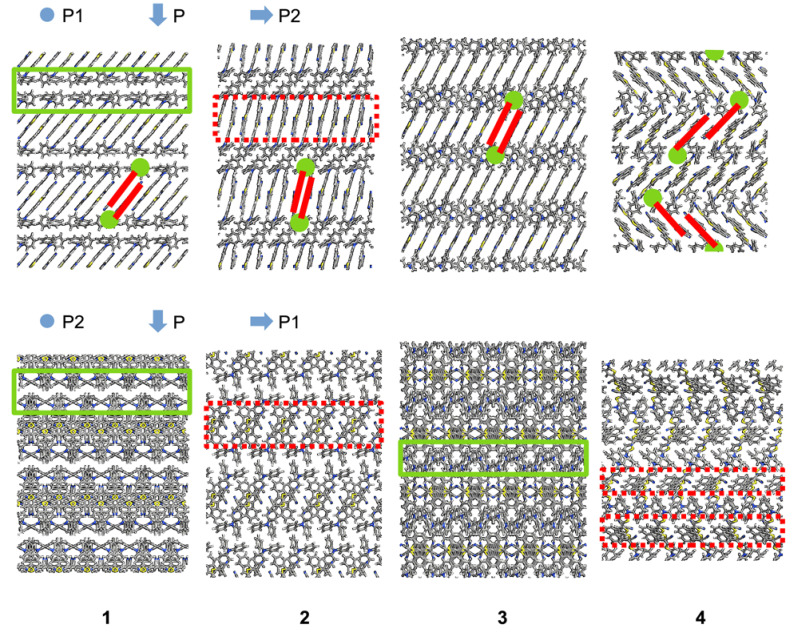
Views of the supercells of compounds **1** through **4** along the directions P1 (**top**) and P2 (**bottom**). The molecules are shown using cylinders to represent bonds. Regions of the cylinders are differently colored to represent carbon, hydrogen, sulfur, and nitrogen using gray, white, yellow, and blue, respectively. The solid (green) and dashed (red) rectangles highlight the phase of the triphenylamine and dicyanovinyl moieties, respectively. Two adjacent molecules are simplified using a line (red) representing the dicyanovinyl and a circle (green) representing the triphenylamine. The crystal of compound **4** is composed of molecules in two different conformations. The transport directions P, P1, and P2 are indicated by arrows, and the circle indicates a direction perpendicular to the page.

## Data Availability

Not applicable.

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
