# Peer review of "Morphology Determines an Efficient Coherent Electron Transport for Push–Pull Organic Semiconductors Based on Triphenylamine and Dicyanovinyl Groups"

_materials, 2023, doi:10.3390/ma16062442_

Round 1

Reviewer 1 Report

Organic materials are being used in various semiconductor devices because of their good charge mobility and tunable properties.

In recent times, there are many detailed studies on Triphenylamine-Based Push−Pull Molecules for Photovoltaic Applications.

Time-dependent DFT studies have also been used to study the ground and excited-state structures.

My question is what makes your study different from others, as there are many comprehensive studies on charge generation that proceeds through the charge transfer state.

Based on only the electronic transmission spectrum, I don’t find this study suitable for publication in the present form.

If authors could provide some significant data and comparative studies to make their work more significant and readable.

Reviewer 2 Report

The manuscript described a theoretical study of a series of organic molecules with push-pull features at the DFT level. The context presented in this manuscript is mostly routine without in-depth discussions. Thus, the whole manuscript reads like a project progress report instead of a thoughtful and informative journal article. Importantly, the charge carrier transport in organic systems is well known to be governed by the coupling of the charge carrier with molecular and lattice vibrations and is highly temperature-dependent. Thus, applying the Landauer-Büttiker formalism, which was designed for ballistic transportation (most likely due to the limitation of the software) to such a complicated situation, is inappropriate. Despite this fundamental problem, the results' presentation is unsatisfactory, especially towards potential charge transport channels in Figure 4, where only ad hoc explanations are presented based on molecular packing. 

Given all these major problems, I would not recommend accepting this manuscript unless significant efforts can be made to reconstruct the work to contain sufficient scientifically rigorous discussions associated with a clear presentation of the data. 

Reviewer 3 Report

From the transport in organic crystals point of view, the article does not cause any complaints. However, in real organic solar cells, as the authors themselves note, "the cell has multiple phases and interfaces", and the morphology of the active layer is stochastic. Accordingly, if the article claims to contribute to the optimization of the characteristics of real solar cells, then it is necessary to clarify (at least qualitatively) how this stochastic nature can be taken into account.

Reviewer 4 Report

materials-1748350

report

Morphology determines an efficient coherent electron transport for push-pull organic semiconductors based on triphenylamine and dicianovinyl groups
Alexander Romero, Jaime Velasco-Medina, Alejandro Ortiz

The articles report the computation of band gap and Fermi energy level of four organic semiconductors compounds containing triphenyl amine, thiophen and dicianovinyl.

The energy spectrum of electron transport for the selected compounds are computed based of their electronic state structure.
The corresponding band gap and the Fermi level on the basis of electronic state structure were calculated.
The electronic state were calculated using:
"1) density-functional based tight-binding method using mio-1-1 parameters...
2) Pulay mixing algorithm with a 10-5 Eh convergence tolerance.
3) fast Fourier-transform based Poisson solver.
4) Fermi-Dirac smearing scheme with a broadening corresponding to 300 K.
The geometry optimization is done using:
1) mesh cut-off of 100 Eh.
2) Monkhorst-Pack grid with k-points density of ca. 10 Å.
3) limited memory Broyden-Fletcher-Goldfarb-Shanno algorithm.
4) maximum force and stress of 80 0.05 eV Å-1 and 0.1 GPa, as convergence parameters."
The simulations were done using the version S-2021.06 of QuantumATK software.

There are steps missing:
-the diagrams of the electronic state for every compound with the valence and conduction bands.
-the correlation between the band gap and wavelength of incident light.
The band gap of 1.26 eV correspounds to near infra-red (NIR) region of 984nm, the band gap of 1.0 eV correspounds to NIR region of 1240 nm,
the band gap of 0.97 eV correspounds to NIR region of 1278 nm,  the band gap of 0.86 eV correspounds to NIR region of 1442 nm.
 How can be this applied to solar cell where the visible electromagnetic wavelength is involved?
From the diagram of electronic transport spectra of all four compounds,
the peaks in the energy range 0.5-1.0eV also correspound to NIR electromagnetic wavelength region: 2480nm-1240nm.
This shows a discrepancy with the aim of this work, the semiconducting solar cell compounds should absorb in the visible range of electromagnetic wave and convert into the energy.

The term "electronic transmission spectrum" should be explained because it shows more a tendency of electronic
 transport as a function of applied energy rather than a electronic transmission into the selected compounds.

Which was the applied energy for computing the "electronic transmission spectrum".
What is the meaning of negative energy in the figures 2,3?

It would be better for reader to shift the supplementary information into the manuscript.

Round 2

Author Response

we found no more commnets to address